# The Lateral Metalation of Isoxazolo[3,4-*d*]pyridazinones towards Hit-to-Lead Development of Selective Positive Modulators of Metabotropic Glutamate Receptors

**DOI:** 10.3390/molecules28196800

**Published:** 2023-09-25

**Authors:** Christina A. Gates, Donald S. Backos, Philip Reigan, Nicholas R. Natale

**Affiliations:** 1Department of Biomedical and Pharmaceutical Sciences, University of Montana, 32 Campus Drive, Missoula, MT 59812, USA; christina.gates@umconnect.umt.edu; 2Skaggs School of Pharmacy and Pharmaceutical Sciences, Anschutz Medical Campus, University of Colorado Denver, Aurora, CO 80045, USA; donald.backos@cuanschutz.edu (D.S.B.); philip.reigan@cuanschutz.edu (P.R.)

**Keywords:** isoxazole, lateral metalation, metabotropic glutamate receptor

## Abstract

Isoxazolo[3,4-*d*] pyridazinones ([3,4-*d*]s) were previously shown to have selective positive modulation at the metabotropic glutamate receptor (mGluR) Subtypes 2 and 4, with no functional cross-reactivity at mGluR_1a_, mGluR_5_, or mGluR_8_. Additional analogs were prepared to access more of the allosteric pocket and achieve higher binding affinity, as suggested by homology modeling. Two different sets of analogs were generated. One uses the fully formed [3,4-*d*] with an N6-aryl with and without halogens. These underwent successful selective lateral metalation and electrophilic quenching (LM&EQ) at the C3 of the isoxazole. In a second set of analogs, a phenyl group was introduced at the C4 position of the [3,4-*d*] ring via a condensation of 4-phenylacetyl-3-ethoxcarbonyl-5-methyl isoxazole with the corresponding hydrazine to generate the 3,4-*d*s **2b** and **2j** to **2n**.

## 1. Introduction

The mGluRs are members in good standing of class C of G-protein-coupled receptors (GPCR). The mGluRs consist of a venus flytrap domain (VFD), which contains the orthosteric glutamate binding site, and the seven transmembrane (7TM) domain, which contains the allosteric site. The mGluRs are located on both post- and pre-synaptic neurons and are involved with signal regulation. Compounds that target mGluRs are important for the treatment of a variety of central nervous system (CNS) disorders, as well as cancer [1,2,3,4,5]. The class is further broken down into three subgroups by sequence homology. Subgroup 1 has the excitatory receptors mGluR_1_ and mGluR_5_. Subgroup 2 (mGluR_2–3_) and Subgroup 3 (mGluR_4,6–8_) are inhibitory [6]. Each has a unique potential therapeutic application associated with it, and therefore ligands with sub-type selectivity are of paramount importance. In particular, mGluR_2_ is a target for treatment of anxiety and schizophrenia, while activation of mGluR_4_ has been postulated to ease the symptoms of Parkinson’s disease and may even slow the progress of the disease [1,2,7,8]. Although there is high sequence homology between the subgroups, the allosteric sites are less conserved and present the most logical opportunity to develop novel small molecules to selectively target them. Using synthetic methods largely developed by Renzi and Dal Piaz [9,10], we reported previously initial hits with [3,4-*d*]s, which exhibited selective activity at mGluR_2_ and mGluR_4_ [11,12]. We next desired to explore the hit-to-lead development of this template to further elaborate this series with liphophilic groups and test the practicality of increasing efficacy and selectivity.

## 2. The Working Hypothesis, Application of Structure-Based Drug Design

Recent crystallographic advances in GPCR structures have allowed for increasing confidence in homology modeling for hypothesis-driven, structure-based drug design.

The allosteric binding site crystal coordinates were used for mGluR_1_, PDB accession number 4OR2 [13]. Ten homology models were compared using the Discovery Studio protocol, and the highest score for goodness of fit was used for the homology comparison. The calculations were performed using Autodock Vina for mGluR_2_ and Discovery Studio for mGluR_4_ [12].

For Discovery Studio, the protein structures were typed with the CHARMm forcefield [14], and energy was minimized with the smart minimizer protocol within Discovery Studio [15] using the Generalized-Born with simple switching implicit solvent model to a root mean square gradient (RMS) convergence of <0.001 kcal/mol prior to use in the docking studies. Docking was performed using the flexible docking protocol [16], which allows for conformational degrees of freedom in both the ligand and the binding site residues. The amino acid residues within the allosteric site were allowed to attain an optimum conformation using flexible docking. For example, the essential unique amino acid binding residues in mGluR_4_ were selected based on Feng’s model of allosteric activation [17] and with numbering of the relative position in the GPCR 7TM based on the protocol established by Isberg [18]: Leu659^3.36^, Met663^3.40^, Leu 753^5.40^, Leu756^5.43^, Leu757^5.44^, Thr794^6.46^, Trp798^6.50^, Phe801^6.53^, Phe805^6.57^, Leu 822^7.32^, and Val826^7.36^.

### 2.1. Computational Prediction Based on the Working Hypothesis for mGluR_2_

Our initial studies indicated that experimentally, the selectivity centered on one particular variable for mGluR_2_, that is, each of the N-Aryl groups contained fluorine substituents (Figure 1) [12]. On closer examination of an expanded ligand set, the most reasonable interactions for the unique non-conserved residues associated with mGluR_2_ were the sulfur-containing residues at the hinge between the Venus flytrap domain and the transmembrane domain. The observation that the program assigned the interaction of the fluorine-containing aryl group with Met728 as a pi-sulfur interaction was especially suggestive. The far intracellular end of the allosteric pocket is formed by Phe776 and Trp 773, which allow just sufficient space to accommodate the geminal di-benzyl substitution.

### 2.2. Computational Prediction Based on the Working Hypothesis for mGLuR_4_

The computational prediction that the unique residue for mGluR_4_, Leu756, is involved directly in binding is encouraging (Figure 2). The fact that the calculation predicts a reasonably strong pi-stacked amide interaction would bode well for subsequent selectivity and affinity. The geminal benzyl groups of **4f** both add important interactions at the VFD and transmembrane domains, with a predicted strong pi-cation interaction with Arg655.

A series of analogs containing a 4-phenyl group were prepared that corresponded to our previous hits at mGluR_2_ and mGluR_4_ [12] from **2j** to **2n**. These contain a rigidly connected lipophilic group and represent a control concerning the role of conformational flexibility at the allosteric site. Compared to our initial analogs in this series [12], the di-substituted analogs had superior predicted CDocker scores to both the mono- and un-substituted analogs. The more rigid 4-phenyl analogs generally showed slightly lower binding energies compared to their 4-methyl analogs.

Caution against confirmation bias should always be accorded to predictions with theoretical calculations, and testing by experiment is always warranted [19,20]. The computational predictions described above, however, (1) suggest a plausible basis for selectivity as the unique sub-type residues are predicted to be involved in the drug receptor interaction and (2) allow for testable hypotheses. Our next task was to prepare the small molecules to test each of these hypotheses.

## 3. Results and Discussion

The synthesis of selectively substituted examples of the [3,4-*d*] scaffold was accomplished in the present study using lateral metalation and electrophilic quenching with non-nucleophilic bases, as summarized in Figure 1 and Table 1 [21,22,23,24,25,26,27].

The Dal Piaz group has pioneered the use of isoxazolyl[3,4-*d*] analogs as a useful scaffold [9,10], both for their intrinsic biological activity [29,30,31,32] and their utility as precursors for medicinally relevant exploration [33,34,35,36,37,38,39,40]. Our rationale for exploring lateral metalation with non-nucleophilic bases lies in previous studies by the Dal Piaz group, in which they reported that upon use of nucleophilic bases for deprotonation, they observed three different routes of rearrangement: isolated compounds **10**, **14**, and **15** (Figure 2) [41,42,43], all predicated on the Kemp elimination [44,45,46,47]. In our hands, we found that extended reflux time evidenced one of these rearrangements, which requires a Kemp elimination, and subsequent rearrangement arising from attack of the nucleophilic base at the pyridazinone carbonyl, loss of carbon dioxide to **12**, followed by protonation and tautomerization to **13**. In our study, we observed the 3-amino-pyrazole **15**, but not its 4-cyano-analog **14** (Figure 2), most probably since we used thermodynamic conditions, whereas the other two documented rearrangements could be kinetic processes.

We therefore turned to lateral metalation and electrophilic quenching (LM&EQ) using non-nucleophilic bases. Originally reported by Micetich [21], we have expanded the application of LM&EQ to functionally complex isoxazoles for a variety of target types [22,23,24,25,26,27].

**Method A** involved direct metalation of the [3,4-*d*] scaffold, conducted in THF, at low temperatures as listed in Table 1. Method A generally accords with a mixture of mono-**3** and di-substitution **4**. These were usually readily separable by chromatography, and the di-substitution product has been confirmed by single crystal X-ray diffractometry of **4a** [11]. In contrast with our previous experience with lateral metalation, [3,4-*d*] proceeds very quickly through mono-electrophile incorporation products **3,** and under almost all conditions studied, it produced a predominant double substitution **4**. Di-substitution can be optimized by using excess bases and electrophiles and is the major product found when the 6-N-aryl group is electron withdrawing, especially in the case of the N-6-3,5-difluorormethyl phenyl, **4f**.

We had previously reported the LM&EQ of isoxazolyl acetals **5**, for which mono-electrophile incorporation was facile, but subsequent di-substitution was found to be very difficult [26,27]. This was found to be the method of choice if predominant mono-substitution was desired. In the case of larger electrophiles (i.e., 1-bromo-methyl-naphthylene **6g**), subsequent hydrazine reactions stopped at the open hydrazone **7g**, and only extended reflux produced the desired closed [3,4-*d*] pyridazinone **3g**. This protection first, mono-LM&EQ, deprotection sequence represents **Method B**. In the 4-phenyl series [3,4-*d*]s **2b** and **2j** to **2n,** it was found that Hunig’s base catalysis served to complete the ring closure efficiently. The characterization of compounds in the 4-phenyl series **2j** to **2n** is given after Appendix A; compound **2b** is previously known [10].

We found that the 3,5-diCl-phenyl example proceeded to produce di-substitution 4f as the major product by method A. We anticipated that metalation between the 3,5-dichloro groups was a distinct alternative pathway (Figure 3), which could result in a benzyne intermediate and subsequent *cine*-substitution [48,49,50,51], followed by addition of the conjugate acid to 18 or 19, and also potential electrophile incorporation to 20 and 21. Therefore, we carefully examined the reaction mixture by HPLC-MS. We did not find evidence of expected products of benzyne intermediates, and the low yield was found to be oligomeric or polymeric baseline material.

The addition of lipophilic groups for compounds **3** and **4** was designed to enhance affinity; however, binding is not pharmacology. The conformational flexibility of the lipophilic additions is intended to enhance lipophilic surface area while simultaneously allowing for the swing of TM6 essential to GPCR activation [3,4,5]. In contrast, series **2b** and **2j** to **2n** contain additional lipophilic character, but in contrast, they are rigid, which would be plausibly expected to alter the pharmacology to inhibit the receptor due to the strut-like lipophilic group acting to prevent the essential movement of the receptor.

## 4. Materials and Methods

Tetrahydrofuran (THF) was dried over activated sieves, then distilled under argon from sodium and benzophenone. Argon gas was passed through tubes with the indicator Drierite for reactions that required an inert atmosphere. NMR spectra were recorded at 400 MHz, unless otherwise specified, in CDCl_3_ solution, and chemical shifts are reported in ppm. The mass spectra were obtained using chemical ionization unless otherwise noted and are reported as *m*/*z* (relative intensity). Starting materials for the lateral metalation were prepared via Dal Piaz’s method for 4-phenyl and 4-methyl 3,4-*d*s [9,10].

All steps were performed in an inert atmosphere, unless otherwise noted. To the pre-dried round bottom cooled under argon, 3,4-*d* was added. After which, dry THF was added in sufficient amounts to reach a concentration of 50 mM. The reaction was then placed and stirred in a cooling bath at the desired reaction temperature or based on solubility for 5 min. Then 1 or 2 eq of the amine base was added via syringe dropwise over 5 min. The reaction mixture was then allowed to react for 30 min, during which time the solution is usually observed to darken. Furthermore, 1 or 2 eq of a cooled 1.7M solution of the electrophile in dry THF (kept at 0 °C or −78 °C) was slowly added dropwise. The solution was allowed to reach room temperature, and a saturated ammonium chloride solution was added at about −20 °C until warming to room temperature. The solvent was evaporated under reduced pressure using a rotary evaporator. The aqueous phase was extracted with dichloromethane (DCM). Moreover, the organic phase was washed with water and brine and finally dried over sodium sulfate overnight. After filtration of the solid, the solution was concentrated using a rotary evaporator and purified by PTLC or column with a mixture of hexanes, ethylacetate, and DCM in a ratio of 6:1:1. The characterization of compounds in the 4-phenyl series **2j** to **2n** is given after Appendix A; compound **2b** is previously known [10].

*4-Methyl-3-phenethyl-6-phenylisoxazolo[3,4-d]pyridazin-7(6H)-one,* **3a**. ^1^H NMR (500 MHz, CDCl_3_): *δ* 7.57 (d, 1H); 7.46 (t, 2H); 7.32 (t, 1H); 7.30 (t, 2H), 7.28 (t, 1H); 7.12 (d, 2H); 3.47 (t, *J* = 7.5 Hz, 2H); 3.18 (t, *J* = 7.5 Hz, 2H), 2.29 (s, 3H). ^13^C NMR: 172.6, 152.9, 152.4, 140.8, 140.0, 138.9, 128.9, 128.9, 128.4, 127.9, 127.1, 125.8, 112.5, 34.3, 29.6, 19.4. C_20_H_17_N_3_O_2_ MW 331.13; ESI-MS *m*/*z* 332.0971 [(M + H)^+^ 90% rel. I.]. HRMS calc’d for C_20_H_18_N_3_O_2_ (M + H^+^): 332.1399, found: 332.1396. −0.9 ppm.*3-(1,3-Diphenylpropan-2-yl)-4-methyl-6-phenylisoxazolo[3,4-d]pyridazin-7(6H)-one,* **4a**. ^1^H NMR (500 MHz, CDCl_3_): *δ* 7.51 (d, 2H); 7.43 (t, 2H); 7.36 (t, 1H); 7.21 (m, 8H); 7.01 (d, 2H); 3.796 (pentet, *J* = 7.5 Hz, 1H); 3.268 (d, *J* = 8 Hz, 4H); 1.86 (s, 3H). ^13^C NMR: 174.5, 152.8, 151.9, 140.6, 139.8, 138.0, 128.8, 128.5, 127.8, 127.1, 125.6, 113.4, 45.0, 40.8, 18.9. C_27_H_23_N_3_O_2_ MW 421.18; ESI-MS *m*/*z* 422.1432 [(M + H^+^), 61% rel. I.]. HRMS calc’d for C_27_H_24_N_3_O_2_ (M + H^+^): 422.1869, found: 422.1871. 0.5 ppm.*3-(2-(4-Chlorophenyl)-2-hydroxyethyl)-4-methyl-6-(p-tolyl)isoxazolo[3,4-d]pyridazin-7(6H)-one,* **3h**. TLC (SiO_2_ 4:4:1 hexane-EtOAc-DCM) R_f_ 0.16. ^1^H NMR (500 MHz, CDCl_3_): *δ* 7.42 (d, *J* = 8.5 Hz, 2H); 7.259 (m, 4H); 7.08 (d, *J* = 8.5 Hz, 2H); 5.1 (br. m., 1H); 4.04 (dd, *J* = 7 Hz); 3.69 (dd, *J* = 15, 7 Hz, 1H); 3.62 (dd, *J* = 7, 15 Hz); 2.40 (s, 3H); 2.386 (s, 3H). C_21_H_18_ClN_3_O_3_ MW 395.1; ESI-MS *m*/*z* 378.0079 [(^35^Cl, M-OH^+^) 47% rel. I.], 379.9991 [(^37^Cl, M-OH^+^) 15% rel. I.].*4-Methyl-3-phenethyl-6-(p-tolyl)isoxazolo[3,4-d]pyridazin-7(6H)-one,* **3c**. ^1^H NMR (400 MHz, CDCl_3_): *δ* 7.29 (d, *J* = 8 Hz, 2H); 7.12 (m, 5H); 6.97 (d, *J* = 8 Hz, 2H); 3.32 (t, *J* = 8 Hz, 2H); 3.03 (t, *J* = 8 Hz, 2H); 2.24 (s, 3H); 2.14 (s, 3H). ^13^C NMR: 172.4, 152.9, 152.4, 139.7, 138.8, 138.2, 137.9, 129.5, 128.9, 128.3, 127.0, 125.6, 112.5, 34.3, 29.6, 21.1, 19.3. C_21_H_19_N_3_O_2_ MW 345.39; ESI-MS *m*/*z* 346.1176 [(M + H)^+^ 100% rel. I.]. HRMS: calc’d for C_21_H_20_N_3_O_2_ (M + H^+^): 346.1556, found: 346.1558. 0.6 ppm.*6-(p-Methoxyphenyl)-4-methyl-3-phenethyl-isoxazolo[3,4-d]pyridazin-7(6H)-one,* **3d**. ^1^H NMR (400 MHz, CDCl_3_): *δ* 7.385 (d, *J* = 8 Hz, 2H); 7.16–7.22 (m, 3H); 7.7.65 (d, J + 8 Hz, d); 6.89 (d, *J* = 8 Hz, 2H); 3.73 (s, 3H); 3.38 (t, *J* = 8 Hz, 2H); 3.11 (t, *J* = 8 Hz, 2H); 2.20 (s, 3H). ^13^C NMR: 172.6, 159.0, 153.0, 152.4, 139.8, 138.9, 133.8, 128.9, 127.1,114.1, 55.6, 34.3, 29.6, 19.4. C_21_H_19_N_3_O_3_ MW: 361.3. HRMS calc’d for C_21_H_20_N_3_O_3_ (M + H^+^): 362.1505, found: 362.1506. 0.3 ppm.*6-(p-Methoxyphenyl)-3-(1,3-diphenylpropan-2-yl)-4-methyl-isoxazolo[3,4-d]pyridazin-7(6H)-one,* **4d**. ^1^H NMR (400 MHz, CDCl_3_): *δ* 7.41 (d, *J* = 8 Hz, 2H); 7.19–7.25 (m, 6H); 7.19 (d, *J* = 8 Hz, 4H); 6.94 (d, *J* = 8 Hz, 2H); 3.83 (s, 3H); 3.81 (pentet, *J* = 8 Hz, 1H); 3.28 (d, *J* = 8 Hz, 4H); 1.85 (s, 3H). ^13^C NMR: 174.5, 159.0, 152.9, 151.9, 139.6, 138.0, 133.7, 128.8, 128.6, 126.9, 114.1, 113.4, 55.6, 45.0, 40.8, 19.0. C_28_H_25_N_3_O_3_ MW: 451.5. HRMS calc’d for C_28_H_26_N_3_O_3_ (M + H^+^): 452.1974, Found: 452.1975. 0.2 ppm.*6-(3,5-Dichlorophenyl)-4-methyl-3-phenethylisoxazolo[3,4-d]pyridazin-7(6H)-one,* **3f**. ^1^H NMR (400 MHz, CDCl_3_): *δ* 7.515 (d, *J* = 4 Hz, 2H); 7.18–7.27 (m, 5H); 7.04 (d, 1H); 3.41 (t, 3 *J* = 8 Hz, 2H); 3.12 (t, 3 *J* = 8 Hz, 2H); 2.2 (s, 3H). ^13^C NMR: 173.0, 152.7, 152.0, 142.1, 140.9, 138.6, 134.8, 128.9, 128.3, 127.2, 124.2, 112.3, 34.2, 29.6, 19.3. C_20_H_15_Cl_2_N_3_O_2_ MW: 400.26; ESI-MS *m*/*z* 400 [(M + H+), 100% rel. I.]; 402 [(M + H) + 2, 67.7]; 404 [(M + H) + 4, 12.2]. HRMS Calc’d for C_20_H_16_Cl_2_N_3_O_2_ 400.0620, Found: 400.0622. 0.5 ppm.*6-(3,5-Dichlorophenyl)-4-methyl-3-(1,3-diphenylpropan-2-yl)-isoxazolo[3,4-d]pyridazin-7(6H)-one,* **4f**. ^1^H NMR (400 MHz, CDCl_3_): *δ* 7.532 (d, *J* = 1.7 Hz, 2H); 7.32 (t, *J* = 1.7,1H); 7.25–7.19 (m, 6H); 7.01–6.99 (d, *J* = 6.4 Hz, 4H); 3.81–3.77 (pentet, *J* = 8 Hz, 1H); 3.30 (s, 2H); 3.28 (s,2H); 1.85 (s, 3H);1.59 (s, 1H). ^13^C NMR: 175.0, 140.7, 137.9, 134.8, 128.8, 128.5, 127.2, 124.1; 45.2, 40.8, 18.9. C_27_H_21_Cl_2_N_3_O_2_ MW: 490.39; HRMS 489 [(M − H)^+^ 100% rel. I.]; 491 (M + H). Calc’d for C_27_H_21_Cl_2_N_3_O_2_ 490.3805, Found: 490.1226.*6-(3,5-Bistrifluoromethylphenyl)-4-methyl-3-(1,3-diphenylpropan-2-yl)-isoxazolo[3,4-d]pyridazin-7(6H)-one,* **4e**. ^1^H NMR (400 MHz, CDCl_3_): *δ* 8.13 (s, 2H);7.82(s, 1H); 7.04 (d, 1H); 7.25–7.18 (m, *J* = 8 Hz, 6H); 7.02–7.00 (d, *J* = 8 Hz, 4H); 3.82–3.79 (m,1H); 3.31 (s 2H); 3.29 (s, 2H); 1.88(s, 3H); 1.6 (s, 1H). C_29_H_21_F_6_N_3_O_2_ MW: 557.49; HRMS *m*/*z* 558 [(M + H)^+^ 100% rel. I.]; 559 [(M + H) + 2]; Calc’d for C_29_H_21_F_6_N_3_O_2_ 557.49, Found: 557.1153.*6-(3,5-Dichlorophenyl)-3-methyl-4-phenylisoxazolo[3,4-d]pyridazin-7(6H)-one,* **2i**. ^1^H NMR (400 MHz, CDCl_3_): *δ* 7. 68 (2H); 7.57 (5H); 7.38 (1H); 5.59 (s, 3H). ^13^C NMR: *δ* 208.2, 194.3, 181.6, 170.9, 164.9, 152.7, 152.0, 142.1, 140.7, 134.8, 134.6, 131.2; 129.3, 127.0, 126.7, 125.6, 124.2, 122.6, 112.5, 31.4, 28.7, 18.9. HRMS Calc’d for C_18_H_11_^35^Cl_2_N_3_O_2_ + H 372.0307, Found: 372.0309. 0.5 ppm.*6-(3,5-Dichlorophenyl)-3-methyl-4-phenyl-6H,7H-[1,2]oxazolo [3,4-d]pyridazin-7-one,* **2j**. ^1^H NMR (400 MHz, CDCl_3_): *δ* 7.57–7.56 (d, 5H);7.37(t, 1H); 2.58 (s, 3H).^13^C NMR 171.0, 152.6, 143.7, 142.2, 134.9, 133.2, 130.5, 129.0, 128.4, 128.0, 124.3, 111.2, 14.0. C_18_H_11_Cl_2_N_3_O_2_ MW: 372.205; HRMS *m*/*z* 372.0309 [(M + H)^+^ 100% rel. I.]; 374.0289 (M + H + 2); Calc’d for C_18_H_11_Cl_2_N_3_O_2_ 372.20538, Found: 372.0309.*6-(4-Methoxyphenyl)-3-methyl-4-phenyl-6H,7H-[1,2]oxazolo [3,4-d]pyridazin-7-one,* **2k**. ^1^H NMR (400 MHz, CDCl_3_): *δ* 7.58–7.53 (t, *J* = 8, 7H); 7.00–6.98 (d, *J* = 8, 2H); 3.85(s, 3H); 2.58 (s, 3H).^13^ C NMR 170.4, 159.0, 152.8, 152.7, 142.6, 133.8, 130.1, 28.8, 128.4, 127.0, 114.0, 111.4, 55.5, 13.9. C_19_H_15_N_3_O_3_ MW: 333.35; HRMS *m*/*z*: 333.1113 [100% rel. I], 334.1147 (20.5%), 335.1181 (2.0%), 334.1084 (1.1%); Calc’d for C_19_H_15_N_3_O_3_ 333.3407, Found: 333.1113.*3-Methyl-6-(4-methylphenyl)-4-phenyl-6H,7H-[1,2]oxazolo [3,4-d]pyridazin-7-one,* **2l**. ^1^H NMR (400 MHz, CDCl_3_): *δ* 7.60–7.58 (m, *J* = 4, 2H); 7.55–7.53(t, *J* = 4, 5H); 2.59 (s, 3H); 2.40 (s, 3H).^13^C NMR 170.6, 152.8, 133.5, 130.3, 128.9, 128.4, 127.7, 115.8, 115.6, 111.4, 14.0. C_19_H_15_N_3_O_2_ MW: 317.35; HRMS Calc’d for C_19_H_15_N_3_O_2_ 317.12, Found: 317.1164.*(E)-Ethyl-4-(1-(2-(3,5-dichlorophenyl)hydrazono)ethyl)-5-phenethylisoxazole-3-carboxylate,* **7f**. ^1^H NMR (400 MHz, CDCl_3_): *δ* 4.34 (q, *J* = 8 Hz, 2H); 3.17 (t, 2H); 2.98 (t, 2H); 1.80 (s, 3H); 1.32 (t, *J* = 8 Hz, 3H). C_22_H_21_Cl_2_N_3_O_3_ MW: 446.3; ESI-MS *m*/*z* 446 [(M + H)^+^, 100% rel. I.], 448 (M + 3^+^, 67.4).*6-(3,5-Dichlorophenyl)-4-methyl-3-phenethylisoxazolo[3,4-d]pyridazin-7(6H)-one,* **3f**. ^1^H NMR (400 MHz, CDCl_3_): *δ* 7.515 (d, *J* = 4 Hz, 2H); 7.18–7.27 (m, 5H); 7.04 (d, 1H); 3.41 (t, 3 *J* = 8 Hz, 2H); 3.12 (t, 3 *J* = 8 Hz, 2H); 2.2 (s, 3H). ^13^C NMR: 173.0, 152.7, 152.0, 142.1, 140.9, 138.6, 134.8, 128.9, 128.3, 127.2, 124.2, 112.3, 34.2, 29.6, 19.3. C_20_H_15_Cl_2_N_3_O_2_ MW: 400.26; ESI-MS *m*/*z* 400 [(M + H)^+^, 100% rel. I.); 402 (M + H+2, 67.7); 404 (M + H + 4, 12.2). HRMS Calc’d for C_20_H_16_Cl_2_N_3_O_2_ 400.0620, Found: 400.0622. 0.5 ppm.*6-(3,5-Dichlorophenyl)-4-methyl-3-(2-(naphthalen-1-yl)ethyl)isoxazolo[3,4-d]pyridazin-7(6H)-one,* **3g**. ^1^H NMR (400 MHz, CDCl_3_): *δ* 7.96 (d, *J* = 8 Hz, 1H); 7.90 (d, *J* = 8 Hz, 1H); 7.79 (d, *J* = 8 Hz, 1H); 7.37–7.61 (m, H); 7.56 (d, *J* = 2 Hz, 2H); 7.35 (t, 1H); 7.35 (d, *J* = 2 Hz, 1H); 7.21 (d, *J* = 8 Hz, 1H), 3.675 (dd, 2H): 3.64 (dd, 2H); 2.00 (s, 3H). ^13^C NMR: *δ* 173.0, 152.7, 152.0, 142.1, 140.7, 134.8, 134.6, 131.2; 129.3, 127.0, 126.7, 125.6. 124.2, 122.6, 112.5, 31.4, 28.7, 18.9. C_24_H_17_Cl_2_N_3_O_2_ MW: 450.32; ESI-MS *m*/*z* 450 [(M + H)^+^,100% rel. I.]; 452 (M + H+2, 68.9); 452 (M + H+4, 13.1). HRMS Calc’d for C_24_H_18_Cl_2_N_3_O_2_ 450.0776, Found: 450.0775. −0.2 ppm.

Kemp Elimination/Rearrangement products, General Experimental: To 25 mL of anhydrous ethanol was added sodium metal (82 mg, 3.5 mol). After the reaction to the alkoxide was complete, 1 mmol of isoxazolo[3,4-*d*] pyridazinone, followed by aldehyde (in a slight excess of 1 mmol), was added in one portion, and the reaction was heated to reflux overnight. The reaction was cooled, the ethanol concentrated by a rotary evaporator, and the residue chromatographed on silica gel using a 4:1:1 hexane-DCM-ethylacetate mixture.

*1-p-Tolyl-3-methyl-(4-p-Chlorocinnamyl)-5-amino-pyrazole,* **15c**. Ar_1_ = p-ClC_6_H_4_; Ar_2_ = p-CH_3_C_6_H_4_; C_20_H_19_ClN_3_O MW 352.8, ESI-MS: *m*/*z* 352.1 (100, M+)); 354.1 (M + 2+, 35).*1-p-Anisyl-3- methyl -(4-p-Chlorocinnamyl)-5-amino-pyrazole,* **15d**. Ar_1_ = p-ClC_6_H_4_; Ar_2_ = p- CH_3_OC_6_H_4_; C_20_H_19_ClN_3_O_2_ MW 368.8, *m*/*z* 368.1 (100, M+); 370.1 (M + 2+, 35).*1-[3,5-Dichlorophenyl]-3-methyl-(4-p-Methoxycinnamyl)-5-amino-pyrazole,* **15f** Ar_1_ = p-CH_3_OC_6_H_4_; Ar_2_ = 3,5-Cl_2_C_6_H_3_; C_20_H_18_Cl_2_N_3_O_2_ MW: 403.28; ESI-MS *m*/*z* 402 (100% rel. I.), 404 (M + 2, 65.5); 406 (M + 4, 11).

## 5. Conclusions

Intense interest in the development of ligands that bind metabotropic glutamate receptors continues [52,53,54,55,56,57,58,59,60], often employing the principles of structure-based hypothesis-driven drug design. The present report is an illustration of how new chemical synthetic methodologies can drive the discovery and exploration of novel biology. In our hands, **Method A** produced di-substituted alkyaltion products directly in the most efficient manner, and **Method B**, while requiring more steps, could selectively give rise to mono-subsitution. With mono- and di-substituted products in hand, we can proceed to further expand the series with double-substituted analogs at the C3 position, as well as explore asymmetric synthesis [61]. We are actively pursuing the synthesis of ligands for glutamate receptors and transporters, as well as their pharmacological evaluation, and will report on our progress in due course.

## 6. Patents

“Isoxazolyl[3,4-*d*] pyridazinones selectively bind metabotropic glutamate receptors”, N.R. Natale, Y.R. Mirzaei, C. Gates, and C. Koerner, Invention Disclosure UM Docket No. UMT-2013-00x, filed on 4 March 2013.

## Data Availability

Computational files are available from the authors.

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
