# Peer review of "The Lateral Metalation of Isoxazolo[3,4-d]pyridazinones towards Hit-to-Lead Development of Selective Positive Modulators of Metabotropic Glutamate Receptors"

_molecules, 2023, doi:10.3390/molecules28196800_

Round 1
Reviewer 1 Report
Instead of "3-carbox-yethyl" should be "3-ethoxycarbonyl".
All acronyms must be explained in the text: PDB, CHARMm forcefield, VFD etc.
The text does not refer to fig. 1 and 2.
The authors must mention which compounds were used as non-nucleophilic bases and as electrophiles. Any base has a nucleophilic character and any nucleophile has a basic character, even if these properties are not as intense, because they depend on both intrinsic and environmental/extrinsic factors.
The title of scheme 2 and the reaction conditions do not appear in the manuscript.
Scheme 2 must be redone. From compound 9 does one reach 12 or the tetrahedral intermediate? when moving from 12 to 13, a proton must be accepted, to compensate for the negative charge.
The text does not mention method B.
Hunig's base catalysis: explanation needed.
Table 32 must be redone explicitly. Yields are expressed in percentages, etc. Are the numbers in parentheses bibliographic indexes?
Scheme 3 has information that must be entered and described/commented in the text, not in the scheme.
Line 183: “(citation)?”
The chemical name begins with a capital letter, e.g. 4-Methyl-3….
The expression "as well as exploring asymmetric synthesis [58]." It cannot be a conclusion because it does not result from the presented study.
My opinion is that tha cap. Results and Dissutions must be redone because the message is difficult to be understood, especially if the reader is a beginner.
Author Response
Response to Reviewers
Reviewer 1
Minor nomenclature issues mentioned have been addressed. The first sentence of review, and fifth from last sentence of review referring to the names in the experimental section)
A list of abbreviations has now been submitted to the editor.
A Legend has been added for Scheme 2. A sentence was added to the description in the text in regards to the protonation and tautomerization. Method B is now mentioned in the text. Scheme 3 has been re-worked and the description now placed in the text (this might have been brought up by reivewer2). Figure 1 and Figure 2 are now specifically called out in the text.
Table 1 (now properly renumbered) has been corrected. The reviewer properly pointed out that brackets are used for citations in this journal and therefore have been replaced with parentheses. I apologize for that confusion. The non-nucleophilic bases are mentioned in the Table, as well as identified in the list of abbreviations. It is appreciated that nucleophilicity of bases is a continuum, however, I believe it is usually accepted that sterically hindered agents such as LDA and LiHMDS are relatively non-nucleophilic compared to strong Bronsted bases such as hydroxide.
Citations concerning the conformational changes of mGluRs subsequent to Kobilka’s Nobel prize in 2012 have been added to line 183
I respectfully disagree with the reviewer on the mention of asymmetric synthesis as our planned next step, in many papers the next stages of the research are usually permitted a brief phrase.
Reviewer 2 Report
Comments
Dear Sir/Mam
After a careful consideration of your manuscript entitled “The lateral metalation of isoxazolo[3,4-d]pyridazinones towards hit-to lead development of selective positive modulators of metabotropic glutamate receptors”, The manuscript should be rejected in the current form.
Major parts of the manuscript submitted was found to possess plagiarised content, which includes methodology and results section. Authors are advised to re-check the manuscript file for plagiarism and bring it to an acceptable range. Since plagiarism is a major parameter that needs to be taken very seriously in research writing, the manuscript is therefore rejected. Also, the plagiarism brings us to an issue of questionable novelty regarding the study. The data includes NMR spectra which is plagiarised; therefore, the novelty of the work should be highlighted.
Other points are listed below:
1. The quality of Figures 1 and 2 can be improved since it is considered the beginning and guide for the results shown below. Figures 1 and 2 are not cited in the text.
2. Authors have not mentioned the results shown in Figures 1 and 2 of which compounds, authors should mention the compound’s name also in the caption of the figures.
3. In the main text table 1 is cited but in the whole manuscript, table 1 is not shown. Similarly, table 2 is shown but not cited in the main text. I think there is some confusion between Table 1 and Table 2 authors should be clear about that.
4. In abstract 2.b compound is mentioned but unfortunately in the whole manuscript, and supplementary file the name of 2.b. and its NMR results are not given.
5. Authors are not mentioned clearly that How did changing the substitution affects the binding efficiency and docking score? Similarly, 4OR2 PDB id is related to the mGlut1 receptors, how authors could relate it with mGluR2 and mGluR4? If authors mentioned the homology modeling word, then described the homology modeling in detail.
6. The working hypothesis related to this work is not clearly discussed in the result and discussion section.
7. Authors should perform any functional assay for the confirmation of activity with the metabotropic glutamate receptors. In line 320, correct the spelling of metabotroic.
8. To elucidate the magnitude of the binding force, MD simulations should be used to represent the complex stability of the synthesized hits with the metabotropic glutamate receptor. Binding free energy should be calculated, and the free energy of amino acid residues in the binding pocket should also be described.
